# Silmitasertib (CX-4945) Disrupts ERα/HSP90 Interaction and Drives Proteolysis through the Disruption of CK2β Function in Breast Cancer Cells

**DOI:** 10.3390/cancers16142501

**Published:** 2024-07-10

**Authors:** Hogyoung Kim, Emma Elkins, Rahib Islam, Bo Cao, Nour Abbes, Kaela Battles, Sihyoung Kim, Sichan Kim, Christopher Williams

**Affiliations:** 1College of Pharmacy, Xavier University of Louisiana, New Orleans, LA 70125, USA; hkim@xula.edu (H.K.); eelkins@xula.edu (E.E.); rislam@lsuhsc.edu (R.I.); bo.cao@lcmchealth.org (B.C.); nabbes@lsuhsc.edu (N.A.); kaelabattles@gmail.com (K.B.); 2College of Arts and Sciences, Xavier University of Louisiana, New Orleans, LA 70125, USA; skim4@xula.edu (S.K.); skim5@xula.edu (S.K.)

**Keywords:** estrogen receptor, ERα36, CK2, HSP90, breast cancer, tamoxifen, SERM

## Abstract

**Simple Summary:**

Estrogen Receptor α (ERα) is a key target for hormonal treatment of breast cancer. However, treatment with antihormonal therapies such as tamoxifen will eventually fail, presumably through hormone-independent signaling through estrogen receptors. It has been shown that a component of this resistance to hormonal therapy may be through expression of an alternative form of estrogen receptor, ERα36. Here, we show that targeting the enzyme CK2, either with the clinical stage CK2 inhibitor CX-4945, or by disrupting expression of its regulatory β subunit, can result in decreased expression of both the canonical form of ERα protein as well as the ERα36 isoform in tamoxifen-sensitive and tamoxifen-resistant breast cancer cells. Our studies provide a rationale for the clinical investigation of CX-4945 in the treatment of ERα-expressing breast cancers.

**Abstract:**

Aberrant estrogen receptor (ERα) signaling mediates detrimental effects of tamoxifen including drug resistance and endometrial hyperplasia. ERα36, an alternative isoform of ERα, contributes to these effects. We have demonstrated that CK2 modulates ERα expression and function in breast cancer (BCa). Here, we assess if CX-4945 (CX), a clinical stage CK2 inhibitor, can disrupt ERα66 and ERα36 signaling in BCa. Using live cell imaging, we assessed the antiproliferative effects of CX in tamoxifen-sensitive and tamoxifen-resistant BCa cells in monolayer and/or spheroid cultures. CX-induced alterations in ERα66 and ERα36 mRNA and protein expression were assessed by RT-PCR and immunoblot. Co-immunoprecipitation was performed to determine the differential interaction of ERα isoforms with HSP90 and CK2 upon CX exposure. CX caused concentration-dependent decreases in proliferation in tamoxifen-sensitive MCF-7 and tamoxifen-resistant MCF-7 Tam1 cells and significantly repressed spheroid growth in 3D models. Additionally, CX caused dramatic decreases in endogenous or exogenously expressed ERα66 and ERα36 protein. Silencing of CK2β, the regulatory subunit of CK2, resulted in destabilization and decreased proliferation, similar to CX. Co-immunoprecipitation demonstrated that ERα66/36 show CK2 dependance for interaction with molecular chaperone HSP90. Our findings show that CK2 functions regulate the protein stability of ERα66 and ERα36 through a mechanism that is dependent on CK2β subunit and HSP90 chaperone function. CX may be a component of a novel therapeutic strategy that targets both tamoxifen-sensitive and tamoxifen-resistant BCa, providing an additional tool to treat ERα-positive BCa.

## 1. Introduction

In 2023, it is estimated that 297,790 new cases of female breast cancer (BCa) were diagnosed, with 43,170 deaths, making BCa the most prevalent newly diagnosed cancer in the United States [1]. More than 80% of BCa tumors express estrogen receptor α (ERα). Consequently, hormonal therapy targeting ERα, either directly with selective estrogen receptor modulators (SERMs) like tamoxifen, pure antagonist (Fulvestrant), or indirectly through the ablation of estradiol synthesis (aromatase inhibitors, gonadotropin analogs), has been an indispensable strategy for patients with ERα(+) disease [2]. Despite the effectiveness of these therapies, primary resistance and the development of refractory disease remains a significant clinical problem impacting patient survival. Various signaling pathways have been demonstrated to play a role in the development of endocrine resistance in BCa. Aberrant ERα signaling may be driven by kinases, which subsequently alter the function of the liganded or unliganded receptor, thereby driving clinical failure of ERα-targeting drugs [3]. As such, it is feasible that the pharmacological inhibition of kinases responsible for aberrant ERα signaling could be an effective therapeutic strategy to treat endocrine-resistant BCa or to enhance the efficacy of combination therapy approaches. Our previous studies have indicated that mRNA expression of the catalytically active α subunit of protein kinase CK2 is associated with decreased relapse-free survival in BCa patients treated with endocrine therapy, and that CK2 impacts the transcriptional activity of ERα in vitro [4]. This suggests the plausibility of CK2 as a contributing factor to endocrine resistance in breast cancer and a rationale for CK2 targeting in ERα(+) BCa.

In addition to crosstalk with various kinase pathways, the incidence of alternatively expressed variant ERα isoforms have been implicated in the development of tamoxifen resistance. ERα36, a 36 kD variant of ERα has garnered interest as a potential driver of endocrine resistance [5,6]. ERα36 is characterized by the loss of the AF-1 domain due to alternative promoter usage while retaining the DNA binding domain and parts of the ligand binding domains [7]. As a result of alternative splicing, the C-terminus of ERα36 is truncated and encodes a unique 27 amino acid C-terminal sequence that mediates downstream signaling with the MAP kinase pathway [5]. Importantly, drugs that function as antagonists to the full-length receptor (ERα66) in the breast, tamoxifen and fulvestrant induce activity of ERα36, which may be involved in mitogenic responses to canonical anti-estrogens [5]. Disrupting the expression or signaling of ERα36 in BCa may be an additional strategy to prevent or treat tamoxifen-resistant BCa.

Given the current understanding of ERα signaling in tamoxifen resistance, an ideal therapeutic strategy to prevent or treat endocrine resistant disease would be to therapeutically target both ERα66 and ERα36 simultaneously. Currently, antiestrogens fail to repress the activity of ERα36 and have been shown to mediate nongenomic, mitogenic responses to typical antiestrogens like tamoxifen and fulvestrant [8]. As such, a pharmacological approach to disrupt both isoforms would likely entail mechanisms other than antagonizing the ligand binding domain. Here, we show that CX-4945 (CX), an orphan drug that targets the oncogenic kinase CK2, represses the proliferation of 4 hydroxytamoxifen (4-OHT)-sensitive parental MCF-7 cells as well as resistant MCF-7 Tam1 cells. CX exposure results in the disruption of both ERα isoforms’ ability to interact with the molecular chaperone HSP90, resulting in accelerated proteolytic degradation of both receptor types. Furthermore, we show that the CK2β subunit is indispensable for ERα stability, providing further evidence of the involvement of CK2 in estrogen signaling. These studies show that CX-mediated inhibition of CK2 activity can serve as a therapeutic strategy to simultaneously disrupt ERα66 and ERα36 signaling in tamoxifen-sensitive and tamoxifen-resistant breast cancer.

## 2. Materials and Methods

### 2.1. Reagents

RPMI 1640 culture medium, penicillin/streptomycin (p/s) solution, and fetal bovine serum (FBS) were all purchased from ThermoFisher Scientific Inc. (Waltham, MA, USA). Silmitasertib (CX-4945), estrogen receptor (ER) antibodies sc-8005 (diluted 1/500), sc-130072 (diluted 1/500), HSP90α/β (diluted 1/500) and GADPH (diluted 1/2000) were sourced from Santa Cruz Biotechnology (Dallas, TX, USA). HSP90 (diluted 1/2000) was sourced from Cell Signaling Technology (Danvers, MA, USA). CK2α (diluted 1/2000), and CK2β (diluted 1/2000) were sourced from Bethyl Laboratories Inc. (Montgomery, TX, USA). Phopho-CK2β (Ser209, diluted 1/1000) was sourced from ThermoFisher Scientific Inc. PCR primers were sourced from Integrated DNA Technologies Inc. (San Jose, CA, USA). Chemiluminescence reagents for immunoblot visualization were purchased from Bio-Rad Laboratories (Hercules, CA, USA). All additional chemicals used in this study were obtained from Sigma Aldrich (St. Louis, MO, USA). Cell culture materials, including culture flasks, were purchased from ThermoFisher Scientific Inc. Obavate media used in the formation of spheroids were obtained from Obatala Sciences (New Orleans, LA, USA).

### 2.2. Cell Culture and Transfection

Tamoxifen-sensitive (MCF-7), tamoxifen-resistant (MCF-7 Tam1), T-47D, MDA-MB-231, and HEK293T were purchased from ATCC (Manassas, VA, USA). The cells were cultured in RPMI 1640 medium supplemented with 10% fetal bovine serum, 2 mM L-glutamine, and 1% penicillin/streptomycin. Moreover, for routine maintenance, each cell line was cultured as a monolayer at 37 °C in a 5% CO_2_, 95% air incubator, and sub-cultured at 80–90% confluency. All cell lines were cultured for 15–20 passages before new stocks were utilized. Cells were placed in phenol red-free RPMI supplemented with 10% charcoal/dextran-treated FBS (CDSS) preceding drug treatment.

Plasmids encoding control Green Fluorescent Protein (pLenti-C-mGFP), ERα66-mGFP, and ERα36-mGFP) were purchased from OriGene Technologies, Inc. (Rockville, MD, USA). HEK293T cells were transfected using Neon electroporation, and subsequently total 293T-mGFP, 293T ERα66-mGFP, and 293T ERα36-mGFP cells were sorted by S3e Cell Sorter (Bio-Rad Laboratories).

### 2.3. siRNA and shRNA-Lentiviral Infection

Control and ERα36 siRNAs as well as control and CK2β shRNAs were purchased from Santa Cruz Biotechnology. MCF-7 and MCF-7 Tam1 cells were transiently transfected with specific siRNAs targeting ERα36 using Lipofectamine 3000, according to the manufacturer’s instructions.

For lentiviral transduction, 3 × 10^4^ MCF-7 and MCF-7 Tam1 cells were seeded in a 6-well cell culture plate. Cells were further incubated for 24 h and then infected by replacing the medium with infection medium containing lentiviral particles and polybrene. The infection medium was removed after overnight incubation and replaced by fresh medium for 2 days. The transduced cells were selected with puromycin dihydrochloride (Santa Cruz Biotechnology), and then treated as described in the figure legends before being subject to total RNA or protein preparation.

### 2.4. Growth and Proliferation Assays

MCF-7, MCF-7 Tam1, and MDA-MB-231 were seeded at a density of 5 × 10^3^ cells/well in 96-well plates. After attachment of the cells, cells were incubated with CX-4945 and tamoxifen. Cell proliferation was monitored for 7 days using the IncuCyte^®^ S3 live cell imager (Sartorius, Ann Arbor, MI, USA). The data analysis was performed by IncuCyte^®^ software (version 2022B Rev2), assessing cell confluence over time. T-47D cells were cultured growth media prior to use in Alamar blue viability assay (ThermoFisher Scientific Inc). Cells were seeded into 96-well plates at 1 × 10^4^ cells/mL (100 μL/well) and incubated for 24 h at 37 °C and 5% CO_2_ to allow the cells to attach. After 24 h, the medium was discarded and the wells thereafter treated with 100 μL of varying concentrations of CX-4945 prepared in the medium. A set of untreated control wells was included in each plate. Following incubation with CX-4945 for 72 h, 10 μL of Alamar blue solution was added to each well. Following 2 h incubation, absorbance was quantified at the wavelengths of 570 nm using SYNERGY H1 (BioTek instruments Inc., Winooski, VT, USA). Each treatment was run in triplicate and each experiment was repeated three times. The percentage cell viability was determined relative to the vehicle-treated control cells.

### 2.5. qRT-PCR

RNA for qRT-PCR was extracted using QIAGEN RNeasy Plus Micro Kit (Valencia, CA, USA). cDNA libraries were constructed from total RNA using the Bio-Rad iScript™ cDNA synthesis kit (Bio-Rad Laboratories). Primers were ordered from Integrated DNA Technologies. The top-down PCR parameters included 30 s at 95 °C, 30 s at 60 °C, and 30 s at 72 °C. Ct values were then compared between groups after being normalized to GAPDH. Samples were run in triplicate. A list of primer sequences used in this study can be found in in Appendix A.

### 2.6. Immunoblotting and Fluorescence Microscopy

Total protein was isolated using RIPA buffer and standard protocols. Protein extracts (15 µg) were utilized for Western blot analyses. Antibodies employed in this assay targeted ERα66 (sc-8005), ERα36 (sc-8005, sc-130072), HSP90, CK2α, p209CK2β, CK2β, β-actin and GAPDH. After treating blots with primary antibody, the appropriate secondary at a dilution of 1/2000 from Jackson ImmunoResearch, Inc (West Grove, PA, USA). was applied. The ChemiDoc imaging system (Bio-Rad Laboratories) was used to image each blot and ImageJ from the NIH was used to quantify.

Fluorescence or brightfield microscopy to assess mGFP, ERα66-mGFP, and ERα36-mGFP expression was performed using EVOS M5000 (ThermoFisher Scientific Inc). Images were merged using Evos M5000 software (version 1.5.1500.493) (Appendix A).

### 2.7. Immunoprecipitation and FLOWPLA (Proximity Ligation Assay)

The effect of protein complex on cells was established using Pierce™ Classic IP Kit (ThermoFisher Scientific Inc.) as previously described [9]. Lysates (1 mg) were precleared by incubation with 20 μL protein A/G plus-agarose beads for 1 h before an overnight incubation at 4 °C with indicated specific antibodies (ERα, HSP90, CK2β) or normal IgG. Protein A/G plus-agarose beads were then added to the lysates, and the mixtures were further incubated on a rotating wheel for 4 h at 4 °C. The beads were pelleted and washed three times in wash buffer. Antibody–antigen complexes bound to the beads were eluted in the sample buffer by boiling and resolved by sodium dodecyl sulfate polyacrylamide gel electrophoresis.

To detect the ERα interaction with HSP90, the Duolink FlowPLA-Deep Red Proximity Ligation Assay (PLA) was used according to the manufacturer’s protocol. Cells were plated onto a 6-well plate 1 day before the experiment and treated the CX for 4 h. Cells were fixed for 15 min with a neutral buffered formalin and treated with 0.1% Triton-X100 in PBS for 5 min. After washing, cell pellets were incubated in blocking solution and immunolabeled (overnight, 4 °C) with primary antibodies, mouse anti-ERα (1:500) and/or rabbit anti-HSP90 (1:1000). Then, the secondary antibodies with attached PLA probes were used in accordance with manufacturer’s instructions. Signals of analyzed complexes were observed in FL-4 using BD Accuri^TM^ plus Flow cytometer.

### 2.8. Statistical Analysis

Using GraphPad Prism, means were compared for samples with ≥3 independent variables, all performed in triplicate. Two-factor analysis of variance followed by significance for all tests was set to 0.05, as previously described.

## 3. Results

CX-4945 impacts the viability of parental and tamoxifen-resistant MCF-7 cells. Since CK2 is overexpressed in various cancer models including breast carcinoma, we analyzed the impact of CK2 inhibition by CX on the viability of the BCa models, MCF-7 and MCF-7 Tam1, using an Incucyte S3 real-time imaging system. Briefly, cells were cultured in 24-well plates with exposure to 2.5, 5 or 10 µM CX, and cell proliferation was monitored as confluency at daily intervals (Figure 1A,C). Similarly, cells were exposed to 4-OHT at 0.25, 0.5 and 1.0 µM to confirm the tamoxifen-sensitive and resistant phenotypes of each model (Figure 1B,D). CX-4945 exposure resulted in concentration-dependent growth inhibition, with greater than 50% reduction in cell confluency at 5 µM on day 4 in MCF-7 and MCF-7 Tam1 cells (Figure 1A,C). In order to more closely mimic the three-dimensional structure of the breast tumors, we assessed the efficacy of CX in inhibiting the growth of tumor spheroid cultures (Figure 1E). Spheroid growth was inhibited in a concentration-dependent manner, closely tracking the efficacy observed in our monolayer studies. These findings agree with several studies that describe the impact of CX-4945 MCF-7 cell viability and proliferation [10]. Because of the observed antiproliferative effects of CX in both tamoxifen-sensitive and resistant models, we investigated the potential additive effects of simultaneous exposure to both CX and 4-hydroxytamoxifen (4-OHT). Our data show that the combination of 4-OHT and CX show an additive effect to inhibit MCF-7 and MCF-7 Tam1 cell proliferation in live cell imaging (Appendix A).

We tested if we could see a similar effect on cell growth of T-47D and MDA-MB-231 (ERα66^−^/36^+^) cells by CX. Growth of MDA-MB-231 cells measured as cell confluence during 7 days was significantly decreased with exposure to 2.5, 5 or 10 µM CX using the Incucyte S3 real time imaging system (Appendix A). Similarly, evidence of decreased cell viability in T-47D cells was observed using Alamar Blue assay (Appendix A). These findings demonstrate that CX has broad antiproliferative capacity in BCa.

ERα66 and ERα36 proteins are overexpressed in tamoxifen-resistant cells and repressed by CX exposure. Our previously published studies suggest that CK2 may play a role in regulating ERα signaling [4,11]. Furthermore, with a pronounced effect of CX on the viability of MCF-7 and MCF-7 Tam1 cells, we assessed the impact of CX on ERα mRNA and protein expression by qRT-PCR and Western immunoblot, respectively. We show that at baseline, MCF-7 Tam1 cells express elevated mRNA (Figure 2A) and protein (Figure 2B) levels of both ERα66 and ERα36 compared to the 4-OHT-sensitive parental MCF-7 cells. To confirm that our pan-ERα antibodies were indeed detecting ERα at 66 kD and 36 kD, we used siRNA directed at the ESR1 transcript, which confirmed the selectivity of the antibody for both ERα isoforms (Figure 2C). In a concentration-dependent manner, CX caused a pronounced reduction in ERα66 and ERα36 mRNA expression (Figure 2D), while also causing downregulation of protein levels in MCF-7, MCF-7 Tam1, MDA-MB-231 (ERα36 only), and T47-D cells (Figure 2E, Figure 2F, Figure 2G, and Figure 2H, respectively). Interestingly, mRNA expression required a minimal concentration of 5 µM CX in MCF-7 cells, whereas protein downregulation was observed at 2.5 µM, demonstrating that the effects of CX on protein levels are at least partially independent of the effects on *ESR1* transcript expression. Under acute exposures, 4-OHT stabilizes ERα expression in MCF-7 cells. 5 µM CX completely reversed 4-OHT-induced overexpression of ERα66 and ERα36 protein (Figure 2I), suggesting that CK2 inhibition might disrupt aberrant ERα signaling associated with the development of anti-estrogen resistance.

Proteolytic degradation contributes to CX-induced ERα66 and ERα36 downregulation in MCF-7 Tam1. Since CX induced a suppression of both mRNA and protein levels of ERα66/36, we investigated whether the changes in protein were due to alterations in protein stability or as a consequence of transcriptional downregulation. In order to ascertain the impact of CX on ERα protein expression in the absence of endogenous ERα, we stably expressed mGFP, ERα66mGFP, or ERα36mGFP in HEK293T cells, under the control of the constitutively active CMV5 promoter (Figure 3A). In this manner, any impact on the *ESR1* promoter elicited by CX would be nullified, thereby demonstrating that the effects of CX on ERα are due to posttranscriptional events. 293T cells were visualized for GFP-expressing ERα66 or ERα36, via fluorescent microscopy (Appendix A). After 24 h of exposure, CX resulted in a concentration-dependent downregulation of ERα66 and ERα36 but not mGFP, suggesting that CX selectively caused proteolytic degradation of both ERα variants (Figure 3B). To determine if the 26S proteosome was responsible for the degradation of ERα variants, we exposed BCa cells cycloheximide (CHX) to reduce de novo ERα translation, and subsequently exposed cultures to CX with or without S26 proteasomal inhibitor MG132 (Figure 3C). Co-administration of MG132 failed to alter protein levels as compared to CX alone, suggesting that CX caused changes in ERα66/36 protein expression in a manner that is primarily independent of the S26 proteosome. This agrees with our preliminary studies, which failed to show a detectable increase in ERα ubiquitination by IP/immunoblot. In MCF-7 Tam1 cells, CX caused a more robust relative diminishment of ERα66 and ERα36 in MCF-7 Tam1 cells, and in the case of ERα66, was partially reversed by MG132. In a similar manner, we observed that basal inhibition of translation (with CHX) or transcription (with actinomycin D) failed to abrogate the effect of CX on ERα downregulation, suggesting that it is indeed due to proteolysis (Appendix A, respectively). These findings suggest that impaired proteolysis of ERα may play a role in tamoxifen resistance, and that CX-mediated inhibition of CK2 activity may provide a pharmacological means to target aberrant ERα expression. Together, these factors demonstrate the CK2 is responsible for maintaining ERα protein expression, and that inhibition of the enzyme results in proteolytic loss of ERα.

The β regulatory subunit of the CK2 holoenzyme is overexpressed in MCF-7 Tam1 cells and is repressed by exposure to CX. To further determine the role of CK2 in mediating resistance to tamoxifen, we measured the levels of mRNA and protein for each of the CK2 subunits by qRT-PCR and western immunoblot in MCF-7 and MCF-7 Tam1 cells. At the mRNA level, CK2α and α’ (the catalytic subunits of the holoenzyme), were upregulated in MCF-7 Tam1 compared to parental MCF-7 cells by 50%, with a much more modest increase in CK2β mRNA expression (Figure 4A). At the protein level, however, only CK2β was strongly overrepresented in MCF-7 Tam1 cells, with an eight-fold increase as compared to parental MCF-7 (Figure 4B). The discordance between mRNA and protein expression suggested that the changes in CK2β expression might also be a consequence of alterations in protein stability of CK2β in tamoxifen resistance. Next, we sought to determine the impact of CK2 inhibition by CX on the expression of CK2 α, α’, and β subunits. CX exposure resulted in a concentration-dependent downregulation of CK2α’ and β subunits but failed to have any significant effects on CK2α protein in either MCF-7 or MCF-7 Tam1 cells (Figure 4C). Loss of S209 phosphorylation of CK2β preceded downregulation of the protein, suggesting that S209 is central autoregulation. Similar to our observations with ERα, decreased CK2α and β levels were not associated with changes in mRNA expression (Figure 4D) at 10 or 24 h time points, providing further evidence that the CX induces protein instability in BCa.

HSP90 is a known substrate for CK2, and ERα is a client protein of the HSP90 chaperone complex [12,13]. As such, we sought to determine if the inhibition of CK2 resulted in perturbation of HSP90-ERα association in breast cancer cells. Following 8 h of exposure to 5 µM CX, we immunoprecipitated both ERα, HSP90, and CK2β, followed by Western immunoblot for ERα, HSP90, or CK2β (Figure 4E). Immunoprecipitation with antibodies against HSP90A/B followed by immunodetection of ERα showed a robust interaction with ERα66 and ERα36, which was strongly disrupted by CX exposure (Figure 4E lanes, 3 and 4). Immunoprecipitation of ERα and immunodetection of HSP90 similarly show that the interaction between ERα and HSP90 is dependent on CK2 activity (Figure 4, lanes 5 and 6). Despite our finding that *CK2β* gene disruption resulted in a loss of ERα66 and ERα36 protein levels, we did not see evidence of a close interaction ERα and CK2β, though CK2β and HSP90 showed an interaction that was not perturbed by CX (Figure 4, lanes 7 and 8). As such, we propose that the CK2 holoenzyme plays a regulatory role in stabilizing ERα/HSP90 interactions, thereby affecting protein stability. To confirm that ERα and HSP90 interactions were disrupted by CX, we used the Flow Proximity Ligation Assay (PLA). MCF-7 cells cultured in E_2_ depleted media served as a control, showing that under basal conditions, ERα interacts with HSP90 (Gray). As expected, CX-exposure results in disruption of the ERα–HSP90 interactions, as evidenced by attenuated fluorescence intensity (Black). Since CX exposure resulted in decreased CK2β protein expression, which paralleled its effects on ERα variants, we explored if silencing of CK2β alone was sufficient to induce a decrease in ERα66/36 expression. Scrambled or CK2β-targeting shRNA plasmids were stably transfected into MCF-7 and MCF-7 Tam1 cells, and the expression of CK2β as well as ERα66/36 was assessed by immunoblot. Knockdown of CK2β resulted in a robust inhibition of ERα66 and ERα36 expression, demonstrating the CK2β regulatory subunit is integral to the stabilization of ERα isoforms in BCa (Figure 4G). In addition to a robust reduction in ERα66 and ERα36 variant levels in both parental and tamoxifen-resistant MCF-7 cell lines, shCK2β cells were approximately 50% less confluent than shRNA control cells, confirming that CK2β contributes to the proliferative capacity of both tamoxifen-sensitive and tamoxifen-resistant MCF-7 cells (Figure 4H).

The CX-triggered ERα66/36 degradation observed in our study has possible clinical applications. It is known that ERα36 is a factor in the progression of several cancers and has been associated with the development of tamoxifen resistance in breast cancer [5,7]. Furthermore, tamoxifen and fulvestrant in ERα66^(−)^ cells induce mitosis, a process that is dependent on ERα36 signaling and subsequent ERK1/2 signaling. Moreover, ERα36 is shown to be the primary mediator of non-genomic estrogen signaling [6,8]. Given this, it stands to reason that CX can be used to target ERα66^(-)^ tumors whose progression is directed by ERα36. Our research demonstrates that, in breast cancers, CK2 signaling is associated with elevated ER expression and tamoxifen resistance, and that pharmacological inhibition with CX can disrupt ERα overexpression and, therefore, cancer cell viability.

## 4. Discussion

Our previous studies revealed that CK2α mRNA expression was a predictive marker for shorter relapse-free survival among patients with ER^(+)^ breast cancer, as well as for those with histological grade 1 or 2 tumors, and among those treated with hormone-based therapy [4]. Additionally, we identified two CK2α phosphorylation sites (S282 and S559) in ERα, identifying the first potential node of an ERα/CK2 signaling axis and suggesting that CK2 inhibition could function to modulate ERα function [12]. Interestingly, mutation of S282 and S559 to phospho-deficient alanine residues resulted in increased estrogen response element-driven Luciferase reporter function, suggesting that phosphorylation of either site may result in suppressed ER activity. In a separate study, phosphorylation of S282 was identified as a positive prognostic marker among breast cancer patients with regard to both recurrence-free survival and overall survival [14]. We propose that CX impacts CK2 mediated functions related to chaperone machinery. HSP90 has been identified as a substrate for CK2 and the phosphorylation of these sites are related to HSP90 client protein affinity [13,14]. Indeed, radicolol and geldamycin, when used to inhibit HSP90, have both been shown to disrupt ERα signaling and protein stability [15,16]. When considered with our own findings, these suggest a potential model in which CK2-mediated phosphorylation of HSP90 (or other targets) promotes HSP90 affinity for ERα, resulting in protection from proteolytic degradation. CX is an orally available drug that has received FDA orphan drug designation for cholangiocarcinoma, medulloblastoma, and biliary tract cancer and has a favorable toxicity profile [17]. CX implementation in BCa may provide an alternative strategy through which selective HSP90 functions may be perturbed indirectly. In ERα breast cancer, it is feasible that this could lead to a clinically relevant disruption of ERα protein expression (Figure 5).

While most our focus here is related to CK2’s function in the ERα axis, it is important to acknowledge that there are several oncogenic pathways that are dependent on CK2, including the PI3K/AKT/mTOR pathway, the NF-κB/IKK pathway, and control of the unfolded protein response. CK2 has also been shown to regulate other aspects of the molecular chaperone activity, namely CDC37, a co-chaperone of HSP90, resulting in the stability of oncogenic kinases, and therefore the proliferation and survival of cancerous cells [18]. Additionally, it has been hypothesized that CK2 protects against the unfolded protein response by activating the XBP1-mediated survival function of IRE1 and inhibiting the eIF2α-mediated promotion of apoptosis [18]. CX downregulates p-s129-AKT, which in turn stabilizes the PDKI-dependent active site Thr308, demonstrating that CK2 is contributes to this central oncogenic pro-survival pathway [19]. Further evidence for the implication of the AKT pathway lies in the link between K63-linked ubiquitination, which promotes the oncogenic activation of AKT, and the resulting pathway [20]. Finally, prosurvival NF-κB phospho-p65 subunit is a substrate for CK2 (p-S529), and is downregulated in the presence of CK2 inhibitor CX-4945, indicating that CK2β propels the IKK/NFκB pathway [18,19].

Our finding that ERα36 degradation is also triggered by CX could have important clinical ramifications. Since its initial discovery in 2005, ERα36 has been implicated in the progression of several cancers and has been associated with the development of tamoxifen resistance in breast cancer [5,7,21]. Additionally, mitogenic effects of tamoxifen and fulvestrant in ERα66^(−)^ cells have been shown to be dependent on ERα36 and subsequent ERK1/2 signaling, and that ERα36 is the primary mediator of non-genomic estrogen signaling [6,8]. Considering this mechanism, it is feasible that CX can be used to target tumors that are ERα66^(−)^, where progression is driven primarily by ERα36. Furthermore, ERα36 expression has been detected in models of ERα66^(−)^ breast cancer, further expanding the potential utilization of CX in breast cancer. Our data support this contention, demonstrating that MDA-MB-231 cells, which only express ERα36, are sensitive to CX, and that ERα36 is downregulated with CX exposure. (Appendix A, respectively). Coupled with these findings that CK2 mediates ERα stability in BCa, it is plausible that CX or otherwise targeting CK2 signaling can impact multiple oncogenic and survival pathways in BCa simultaneously and becoming an effective alternative for the treatment of BCa.

Our studies show that CK2 signaling in breast cancer is associated with elevated ER expression and tamoxifen resistance, and that pharmacological inhibition with CX can perturb ERα overexpression and cancer cell viability.

Our findings here bring to the questions several important lines of inquiry yet to be resolved regarding the role of the CK2:ERα signaling axis in BCa. Among the questions is whether or not the effects of CX observed in cells will be recapitulated in vivo. Our spheroid models suggest efficacy in a 3-dimensional context, but lack the cellular heterogeneity found in tumors. In addition, the pharmacokinetic parameters governing therapeutic efficacy are lacking in monolayer and 3D models. As such, our future work will extend these studies into pre-clinical models of BCa. Interestingly, we did not detect CK2β in complexes with HSP90, despite its necessity for ERα stabilization. There are several potential explanations for this observation: (1) CK2β may be potentiating the activity of an as-yet-unknown ERα interactant. (2) The stoichiometry of the pool of CK2β and ERα could be such that it prevents the detection of the complexes by immunoprecipitation. If CK2β:ERα complexes exist, they may exist in a small pool of the available milieu. (3) The interactions could be weak or transient, or (4) methodologically, the CK2β:ERα containing complexes may interact in a manner such that it precludes antibody binding. In order to address this ambiguity, we will perform studies with antibody-free binding assays, as well as proximity assays which can capture transient interactions. Determining the precise regulatory events which drive the dissociation of HSP90 and ERα will be important both mechanistically, and with regard to optimizing the potential clinical utilization of CX or other CK2 targeting strategies. Similar strategies of combining anti-estrogenic therapies have proven effective, exemplified by combination regimens of endocrine therapy with CDK4/6 inhibitors. Given safety profile CX, could be another component of the armamentarium for BCa treatment.

## 5. Conclusions

Here, we show that CK2 activity is integral to sustained ERα66 and ER36 protein stability by regulating its interactions with HSP90. These findings may provide a rationale for the study of CK2 inhibitor CX as a combination therapy in the hormonal treatment of BCa.

## Figures and Tables

**Figure 1 cancers-16-02501-f001:**
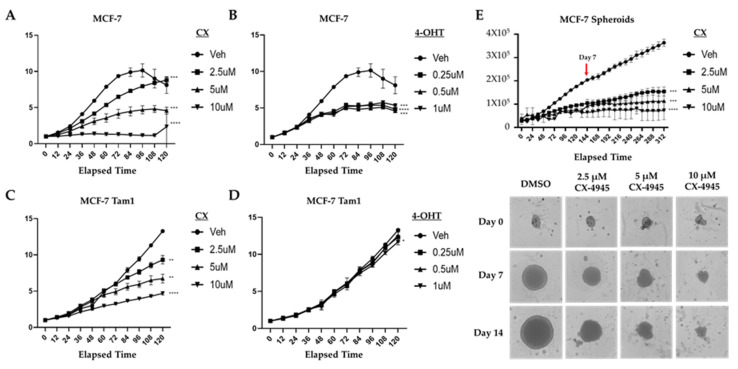
Temporal and concentration-dependent effects of CX on BCa cells. Live cell imaging of cell confluence with exposure of MCF-7 (**A**,**B**) to CX or 4-OHT, respectively. MCF-7 Tam1 cells (**C**,**D**) were treated as described in (**A**,**B**). (**E**) MCF-7 tumor spheroids were exposed to CX for up to 15 days and spheroid diameter monitored (scale bar: 300 μm). Statistical analysis was performed by 2-way ANOVA or mixed effects, using Dunnett’s multiple comparison’s post hoc test, where * *p* ≤ 0.05, ** *p* ≤ 0.01, *** *p* ≤ 0.005, **** *p* ≤ 0.001 and compared with the control group.

**Figure 2 cancers-16-02501-f002:**
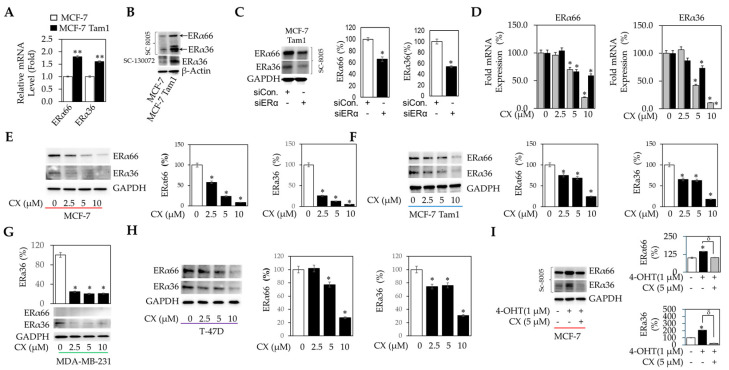
Effects of CX on ERα66 and ERα36 expression in tamoxifen-sensitive and -resistant MCF-7 cells. (**A**) ERα66 and ERα36 RNA transcript levels in MCF-7 and MCF-7 Tam1 cells were measured by qRT-PCR. (**B**) ERα66 and ERα36 protein levels in MCF-7 and MCF-7 Tam1 cells were determined by western immunoblot. (**C**) MCF-7 Tam1 cells were transfected with a scrambled or ERα36-targeting siRNA, and subsequently assessed ERα66/36 protein levels by immunoblot. (**D**) ERα66 and ERα36 RNA transcript levels in MCF-7 and MCF-7 Tam1 cells were measured by qRT-PCR, following 24 h CX exposure at the indicated doses. ERα66 and/or ERα36 protein levels in MCF-7 (**E**), MCF-7 Tam1 (**F**), MDA-MB-231 (ERα36 only, (**G**)) and T47-D (**H**) cells by Western immunoblot following 24 h CX exposure at the indicated concentrations. (**I**) The impact of CX on the expression of ER66/36 levels in the presence and absence of 4-OHT in MCF-7 cells by immunoblot. Ratiometric analysis was performed to assess ERα66/36 changes relative to control and normalized to β-actin expression. For qRT-PCR, statistical analysis was performed by 2-way ANOVA using Dunnett’s multiple comparison’s post hoc test, where * *p* ≤ 0.05 and ** *p* ≤ 0.01 compared with the control group. Full blots are presented in Appendix A.

**Figure 3 cancers-16-02501-f003:**
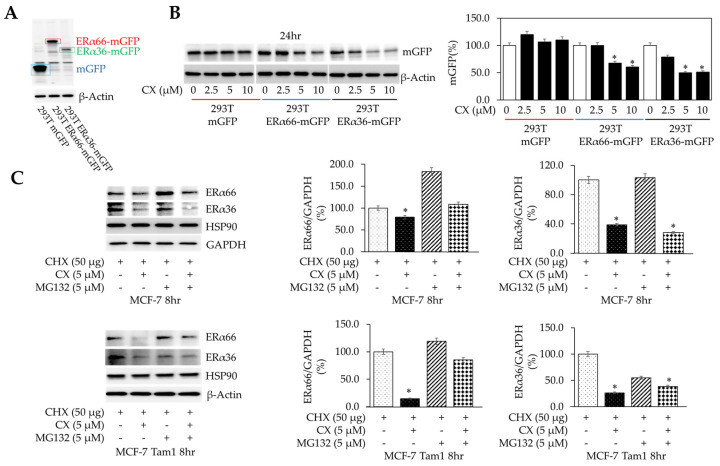
Role of proteolysis in CX-mediated downregulation of ERα66/36. (**A**) HEK293T cells stably transfected with mGFP, ERα66-mGFP, or ERα36-mGFP. Immunoblot using mGFP antibodies were performed. The panel shows mGFP immunoreactivity of each cell line. (**B**) 293T-mGFP, 293T-ERα66-mGFP, and 293T-ERα36-mGFP cells were exposed to CX for 24hr. mGFP and β-actin expression were ascertained by immunoblot. (**C**) MCF-7 and MCF-7 Tam1 cells were exposed to CX in the presence or absence of protein synthesis inhibitor cycloheximide (CHX) and proteosome inhibitor MG132. ERα66/36, HSP90, GAPDH, and β-actin expression were ascertained by immunoblot. For WB, statistical analysis was performed by 2-way ANOVA using Dunnett’s multiple comparison’s post hoc test, where * *p* ≤ 0.05 and compared with the control group. Full blots are presented in Appendix A.

**Figure 4 cancers-16-02501-f004:**
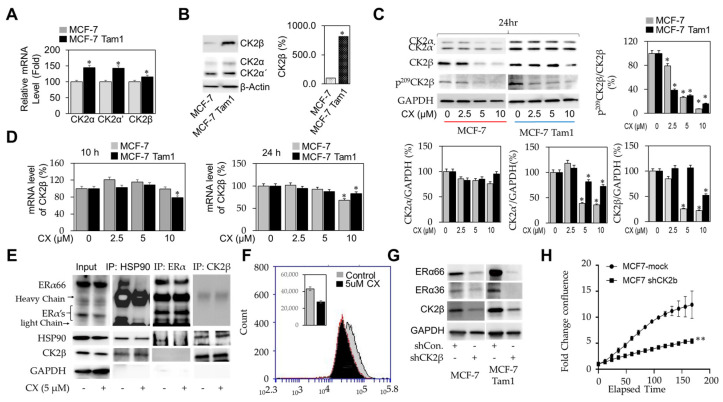
CX exposure impacts CK2β subunit phosphorylation and expression. (**A**) Expression of CK2 subunits (α, α’, β) in MCF-7 and MCF-7 Tam1 by qRT-PCR. (**B**) α, α’, and β subunit protein levels in MCF-7 and MCF-7 Tam1 cells were determined by immunoblot, and β-actin. (**C**) α, α’, β subunit, and S209 phosphorylation of CK2β protein level in MCF-7 and MCF-7 Tam1 cells with 5 µM CX by western immunoblot following 24 h. (**D**) of β subunit expression in MCF-7 and MCF-7 Tam1 with CX by qRT-PCR following 10 or 24 h. (**E**) Co-immunoprecipitation of HSP90, ERα, and CK2β in MCF-7 Tam1 cells after 4 h exposure to 5 µM CX. (**F**) ERα/HSP90 signal measured by flow-PLA for MCF-7 cells treated with CX. The stimulated cells without the corresponding target binding probes served as the control. Data represent of 3 independent experiments. (**G**) MCF-7 and MCF-7 Tam1 cells were treated with control or CK2β shRNA. The transduced cells were selected with puromycin dihydrochloride. Protein lysates were analyzed for ERα66, ERα36, and CK2β, expression by Western blotting. (**H**) Incucyte^®^ proliferation assay analysis of MCF-7 cells seeded at 5 × 10^3^ cells/well in a 96-well plate. The cells were transfected with a shRNA control or targeting CK2β. For qRT-PCR and WB, statistical analysis was performed by 2-way ANOVA using Dunnett’s multiple comparison’s post hoc test, where * *p* ≤ 0.05 and ** *p* ≤ 0.01 compared with the control group. Full blots are presented in Appendix A.

**Figure 5 cancers-16-02501-f005:**
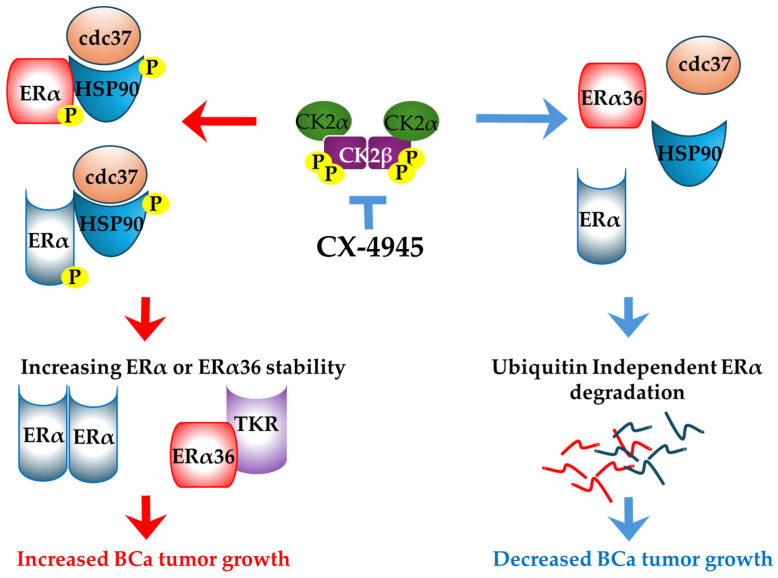
Model of CK2 regulated ERα stability in BCa. Auto-phosphorylated CK2 phosphorylates ERα66, ERα36, and HSP90, resulting in increased association and stability of the estrogen receptors. Inhibition by CX-4945 results in reduced phosphorylation of the substrates, resulting in enhanced degradation of both ERα isoforms.

## Data Availability

Data will be made available on request.

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
