# Peer review of "Silmitasertib (CX-4945) Disrupts ERα/HSP90 Interaction and Drives Proteolysis through the Disruption of CK2β Function in Breast Cancer Cells"

_cancers, 2024, doi:10.3390/cancers16142501_

Round 1
Reviewer 1 Report (Previous Reviewer 1)
Comments and Suggestions for Authors
Although the author has revised this manuscript based on my previous comments, I still have the following small suggestions:
1. Please provide the scale abr for the microscope and the ordinate units for the statisticsfor figure 1E.
2. The "P" representing statistical significance should be in italics.
3. Please add statistical analysis to all WB Figures in Figures 1-4.
Comments on the Quality of English Language
Minor editing of English language required.
Author Response
Please see the attachment.

This manuscript is a resubmission of an earlier submission. The following is a list of the peer review reports and author responses from that submission.
Round 1
Reviewer 1 Report
Comments and Suggestions for Authors
-
This study investigates the impact of CX-4945 (CX), a CK2 inhibitor, on ERα66 and ERα36 signaling in breast cancer cells. The researchers demonstrate that CX suppresses proliferation and spheroid growth in tamoxifen-sensitive MCF-7 and MCF-7 Tam1 cells. CX treatment also leads to a significant decrease in both endogenous and exogenously expressed ERα66 and ERα36 protein. Silencing of CK2β, a regulatory subunit of CK2, shows similar effects as CX treatment. Co-immunoprecipitation experiments reveal that the interaction between ERα66/36 and molecular chaperone HSP90 is CK2-dependent. These findings suggest that targeting CK2 with CX may offer a promising therapeutic approach for treating ERα-positive breast cancer, including both tamoxifen-sensitive and tamoxifen-resistant cases. This is an interesting mauscript, but I have several following concerns:
1. After describing the phenomenon in the section "CX-4945 impacts viability of parental and tamoxifen-resistant MCF-7 cells," please specify the specific part in Figure 1 that corresponds to this observation.
2. Since CX has been shown to decrease the mRNA expression of ERα66 and ERα36, it is important to investigate whether the reduction in protein expression of ERα66 and ERα36 by CX is partially due to transcriptional inhibition.
3. Can you please provide the corresponding conclusions and figures related to the sentence "We investigated the potential additive effects of simultaneous exposure to both CX and 4-hydroxytamoxifen (4-OHT)"?
4. In Figure 2A, please clarify which form represents ERα66 and ERα36.
5. In Figure 2D, please indicate which columns represent the sensitive and resistant types of MCF-7 cells to tamoxifen.
6. In Figure 4C, please adjust the position of "CK2α'" appropriately.
7. What is the distinction between WT-ERα and ER66, and can they be used interchangeably?
8. Regarding Figure 7, specifically lanes 5 and 6, could you please explain their significance in relation to the immunoprecipitation of ERα and subsequent immunodetection?
Minor editing of English language required
Reviewer 2 Report
Comments and Suggestions for Authors
The paper investigates the role of CK2 in regulating estrogen receptor alpha (ERα) signaling and its potential as a therapeutic target in breast cancer treatment. The study demonstrates that inhibiting CK2 with CX-4945 can decrease the expression of both ERα66 and its variant ERα36, providing a rationale for its clinical use in combating tamoxifen-resistant breast cancer. The results utilize various experimental approaches including live cell imaging, qRT-PCR, and immunoblotting to validate the findings. However, the text is repetitive in some sections and the figures are mislabeled in many parts, which might obscure the key findings. Additionally, while the mechanistic insights into CK2's interaction with ERα and HSP90 are valuable, a more streamlined presentation could enhance readability and impact. Several aspects warrant critical evaluation:
1. The evidence linking CK2 activity directly to the stabilization of ERα via HSP90 is compelling but could benefit from additional supporting experiments. For instance, further validation through knockdown studies of HSP90 in the presence of CX-4945 would strengthen the claim that CK2 regulates ERα stability primarily through HSP90 chaperone function.
2. The use of MCF-7 and MCF-7 Tam1 cells is appropriate for modeling tamoxifen-sensitive and resistant breast cancer. However, the study could be strengthened by including additional cell lines, such as primary breast cancer cells or other ERα-positive and ERα-negative lines, to validate the generality of the findings.
3. The incorporation of 3D spheroid models is a significant strength, as it better mimics the tumor microenvironment compared to monolayer cultures. Yet, the study could benefit from discussing any limitations of these models and how they might impact the interpretation of the results.
4. In the text, the MCF-7 and MCF-7 Tam1 cells are referenced as shown in Figure 1A, top two panels, but this is not accurately reflected in the figure section.
5. The text does not refer to the figure number showing spheroid growth.
6. The description of which model is tamoxifen-sensitive versus tamoxifen-resistant needs to be clearer.
7. Larger and clearer images of the spheroids in Figure 1E would improve visibility.
8. In the figure legend for Figure 1, the word “temporal” needs to have a capital “T.”
9. There is confusion and mislabeling of subfigures A, B, C, and D in Figure 1.
10. What is the rationale behind treating spheroids with CX for up to 15 days? How was this time point chosen?
11. In Figure 2A, what do each of the bar groups represent? The x-axis lacks labels for ERα66 and ERα36.
12. In Figure 2D, what do the gray and black bars represent?
13. The *p-value for the qRT-PCR analysis in Figure 2D is compared to what?
14. There is mislabeling of subfigures A, B, and C in Figure 3 as well.
15. Why were all samples not run on the same gel? Running all samples on a single gel would facilitate easier comparison of exposure levels for each protein in each group in the immunoprecipitation and immunoblot assays.
16. Subfigure C is mentioned twice in Figure 4, referring to both the western blot and qRT-PCR.
17. The following phrase is repetitive in both the results and discussion sections and needs to be revised: “Our finding that ERα36 degradation is also triggered by CX could have important clinical ramifications. Since its initial discovery in 2005, ERα36 has been implicated in the progression of several cancers and has been associated with the development of tamoxifen resistance in breast cancer [7] [20] [8]. Additionally, mitogenic effects of tamoxifen and fulvestrant in ERα66(-) cells have been shown to be dependent on ERα36 and subsequent ERK1/2 signaling, and that ERα36 is the primary mediator of non-genomic estrogen signaling [6] [9]. Considering this mechanism, it is feasible that CX can be used to target tumors that are ERα66(-), where progression is driven primarily by ERα36. Our studies show that CK2 signaling in breast cancer is associated with elevated ER expression and tamoxifen resistance, and that pharmacological inhibition with CX can perturb ERα overexpression and cancer cell viability.”
18. The manuscript could explore potential alternative pathways or mechanisms through which CK2 might influence ERα stability and function. Discuss the limitations of the study, future research directions, and the significance of the findings. This would provide a more comprehensive understanding and identify any other contributing factors.
Overall, this study presents a promising therapeutic strategy targeting CK2 to overcome tamoxifen resistance in breast cancer. However, a more streamlined presentation of the mechanistic pathway, inclusion of additional experimental models, and validation would significantly enhance the manuscript’s impact and credibility.
Comments on the Quality of English Language
The text is repetitive in some sections and the figures are mislabeled in many parts, which might obscure the key findings
Reviewer 3 Report
Comments and Suggestions for Authors
The author attempted to disrupt the interaction between ERα and HSP90 by targeting CK2β. However, additional experiments are necessary to confirm these findings.
Minor comments:
1. The molecular weight of the protein should be included in all the Western blots.
2. The bar graph results for Western blot and PCR should be uniform across all figures.
3. The protein concentration used for immunoblotting and immunoprecipitation is missing.
4. Antibody dilutions should be provided to enable readers to reproduce the experiments.
Major comments:
1. What is the rationale for switching from RPMI to DMEM medium for the treatment?
2. Why did the author choose MCF-7 cell lines over T-47D cell lines for the CX-4945 treatment? According to the Human Protein Atlas, T-47D cells have higher CK2 expression than MCF-7 cells.
3. The author should confirm cell viability using dyes such as YOYO1 and NucRed to further verify the concentration-dependent effects with the S3 live-cell imager. The built-in AI program cannot provide 100% accuracy.
4. How did the author perform tumor spheroid cell viability analysis?
5. Why did the author use different concentrations of 4-OHT for cell viability and Western blot assays?
6. CK2 protein expression was not included in the Western blot analysis for MCF-7, MCF-7 Tam1, and CX treatment samples.
7. Quantification of Western blots should be included.
8. The 10 µM CX treatment showed a reduction of both wild-type and ERα36 as determined by PCR. Therefore, the author should have used 10 µM CX treatment instead of 5 µM in Figure 2G.
9. Why are there so many signals in the mGFP constructs?
10. In Figure 3A, for the transfection of mGFP, ERα66-mGFP, or ERα36-mGFP, there was no treatment with CX, but the figure legends indicated otherwise.
11. The images in Figure 3 do not match the descriptions in their legends.
12. The Western blot for ERα36 needs improvement. In some images, there is a single band, while in others, there are too many bands for the protein expression.
13. In Figure 4B, the lower band signal intensity for CK2α is higher than the upper band, whereas in Figure 4C, the lower band signal intensity is less than the upper band. How did this discrepancy occur even though the same cell lines were used?
14. Why did the author change from a 24-hour CX treatment to a 4-hour treatment in Figure 4E?
15. How did the 5 μM CX treatment completely abolish the expression of HSP90 in Figure 4E after 4 hours compared to Figure 3C after 24 hours?
16. The Co-IP results do not clearly indicate where these interactions occur. The author needs to confirm these interactions with a PLA experiment to determine if they occur in the cytoplasm or nucleus.
17. There are conflicts between the images and their corresponding legends. The author needs to verify all the images.
18. Further confirmation in vivo is mandatory for a better understanding.
19. What was the reason for the absence of CK2β under IP ERα?
20. The statistical analysis needs to be verified
Comments on the Quality of English Language
Moderate editing is required
Reviewer 4 Report
Comments and Suggestions for Authors
The manuscript "Silmitasertib (CX-4945) disrupts ERα/HSP90 interaction and drives proteolysis through disruption of CK2β function in breast cancer cells" is a worthy example of this kind of research. The text is written in a clear and concise manner. The conclusions of the paper seem to be reasonable and they are supported by a large amount of data. The manuscript makes a favourable impression.
Perhaps the illustrations should be reworked to make them more symmetrical. In addition there are minor spelling errors in the text, such as in the legend of figure 1.
